# Overlooked but Serious Gallbladder Disease during Extracorporeal Membrane Oxygenation: A Retrospective Analysis

**DOI:** 10.3390/jcm11082199

**Published:** 2022-04-14

**Authors:** Hee Young Kim, Jin Wook Jang, Hye-Jin Kim, Woo Hyun Cho, Mihyang Ha, Bong Soo Son, Hye Ju Yeo

**Affiliations:** 1Department of Anesthesia and Pain Medicine, Pusan National University Yangsan Hospital, Yangsan 50612, Korea; yuvi1981@naver.com (H.Y.K.); west082@hanmail.net (H.-J.K.); 2Department of Anesthesia and Pain Medicine, School of Medicine, Pusan National University, Yangsan 50612, Korea; 3Research Institute for Convergence of Biomedical Science and Technology, Pusan National University Yangsan Hospital, Yangsan 50612, Korea; popeyes0212@hanmail.net; 4Division of Allergy, Pulmonary and Critical Care Medicine, Department of Internal Medicine, Pusan National University Yangsan Hospital, Yangsan 50612, Korea; a_warm_mind@naver.com; 5Department of Internal Medicine, School of Medicine, Pusan National University, Yangsan 50612, Korea; 6Interdisciplinary Program of Genomic Science, Pusan National University, Yangsan 50612, Korea; mh2059389@naver.com; 7Department of Thoracic and Cardiovascular Surgery, Pusan National University Yangsan Hospital, Yangsan 50612, Korea

**Keywords:** cholecystitis, ECMO, gallbladder, gallstone, mortality, plasma hemoglobin

## Abstract

Background: To date, there have been no reports assessing the incidence, risk factors, and clinical outcomes of GB disease in patients receiving ECMO for cardiorespiratory failure. Methods: The medical records of adults (aged > 18 years) who underwent ECMO between May 2010 and October 2019 were retrospectively reviewed. We investigated the prevalence and related factors of GB disease during ECMO therapy, compared clinical outcomes between patients with and without GB disease, and performed propensity-matched analysis. Results: In total, 446 patients were included, and symptomatic GB disease was found in 62 patients (13.9%, 76.2/1000 ECMO days). Complicated GB disease occurred in 42 patients (9.4%, 89.4/1000 ECMO days) and presented as acute cholecystitis, acute cholangitis, and biliary pancreatitis in 33 (7.4%), 7 (1.6%), and 5 (1.1%) patients, respectively. In multivariate Cox regression analysis, longer ECMO support (>2 weeks) (hazard ratio (HR), 2.95; 95% confidence interval (CI), 1.69–5.15) and elevated plasma hemoglobin (Hb, >50 mg/dL) (HR. 2.12; 95% CI, 1.18–3.78) were significantly associated with the development of GB disease. In the propensity-matched cohort, the intensive care unit (ICU) and hospital survival rates were significantly lower for patients with GB disease than for those without GB disease (ICU survival rate, 64.5% vs. 84.7%; hospital survival rate, 59.7% vs. 81.5%). Conclusion: The incidence of GB disease was higher in patients who received ECMO than in the general ICU patients. Furthermore, elevated plasma Hb and prolonged ECMO therapy were significant factors for the development of GB disease during ECMO therapy.

## 1. Introduction

Extracorporeal membrane oxygenation (ECMO) is an important life-saving option for the management of severe refractory heart and lung dysfunction. Improvements in ECMO equipment and increased experience have made it possible to use ECMO for extended periods [1]. However, complications following the long-term use of ECMO are very common, and mortality and morbidity remain high [2,3,4,5,6].

The development of complicated gallbladder (GB) diseases such as cholecystitis in critically ill patients is now a well-recognized complication and may occur as a result of treatment [7]. It can also progress to multiple organ failure and is associated with a prolonged duration of intensive care unit (ICU) stay and high mortality [8]. The pathogenesis of complicated GB disease is multifactorial [9]. Patients treated with ECMO are the most critically ill patients and often have other risk factors in addition to those commonly associated with being critically ill. Patients who receive ECMO may develop complicated GB disease secondary to long-term fasting, total intravenous nutrition, and diuretic therapy. Furthermore, shear stress, generated by blood flow through the ECMO circuit and oxygenator, can cause ECMO-induced hemolysis, which can also lead to complicated GB disease [10].

Although GB disease is a serious complication in critically ill patients, its importance in patients receiving ECMO has so far been overlooked. To date, no reports have been published regarding the incidence, risk factors, and clinical outcomes of GB disease in ECMO patients. In this study, we investigated the prevalence, related factors, and clinical outcomes of GB disease during ECMO support.

## 2. Methods

### 2.1. Study Population

The medical records of adults (aged > 18 years) who underwent ECMO at Pusan National University Yangsan Hospital from 1 May 2010 to 30 October 2019 were retrospectively reviewed. The study was approved by the Institutional Review Board of Pusan National University Yangsan Hospital (05-2021-095), and the need for informed consent was waived due to the retrospective nature of the study. Patients who received ECMO for less than 48 h were excluded from the analysis. The included patients were classified as those with GB disease (G group), which included symptomatic GB disease, and those without GB disease (N group). GB disease was diagnosed based on ultrasound and/or computed tomography (CT) findings such as necrosis and gangrene or distension and thickening of the GB, clinical symptoms, and deteriorating multiple organ dysfunction in the absence of other suspicious foci. We performed abdominal ultrasonography or abdominal CT in patients supported with ECMO who presented with one of the following symptoms: elevation of liver function tests (LFT), septic shock, abdominal pain, nausea, vomiting, or ileus. Gallstone disease was defined as having symptomatic gallstones. Patients with gallstone disease were further divided into those with uncomplicated GB disease and those with complicated GB disease. Uncomplicated gallstone disease was defined as having biliary colic in the absence of gallstone-related complications. Complicated gallstone disease was defined as having the presence of gallstone-related complications, including acute cholecystitis, cholangitis, gallstone pancreatitis, gallstone ileus, and Mirizzi syndrome [10,11]. Patient demographics, renal replacement therapy usage, main diagnosis, and laboratory data were collected retrospectively.

### 2.2. Statistical Analyses

Continuous variables were examined for normality using the Shapiro–Wilk test. Normally distributed variables were compared using the Student’s *t*-test, whereas non-normally distributed variables were compared using the Kruskal–Wallis test. Numerical results are expressed as mean ± standard deviation (SD) or median (interquartile range) as appropriate. Categorical variables were examined using Fisher’s exact test or a chi-squared test. Statistical significance was defined as *p*-value < 0.05. To evaluate the risk factors associated with GB diseases in patients receiving ECMO, a Cox proportional hazards model was used. All potential clinical factors were evaluated using univariate Cox regression analysis, and multivariate Cox regression analysis was conducted for variables with a *p*-value < 0.05. Furthermore, the risk of GB disease according to ECMO duration or plasma hemoglobin (Hb) level was expressed as a figure using a forest plot. To improve the readability of the figure, the odds ratio (OR) forest plot was plotted as log OR. In the comparison of the clinical outcomes between the two groups, to reduce bias caused by confounding variables, we conducted propensity score matching analyses (1:2), which were adjusted for age and severity. All the statistical analyses were performed using IBM SPSS Statistics (version 25.0; IBM Corp., Armonk, NY, USA) or R 3.6.2.

## 3. Results

### 3.1. Incidence and Clinical Features of Symptomatic GB Disease during ECMO Therapy

During the study period, 549 patients underwent ECMO. Of these, 103 patients who underwent ECMO for less than 48 h were excluded from the analysis, and a total of 446 patients were included in this study (Figure 1). Symptomatic GB disease was present in 62 patients (13.9%) at a rate of 76.2 per 1000 ECMO days. Uncomplicated GB disease occurred in 20 patients (4.5%) at a rate of 58.1 per 1000 ECMO days. Complicated GB disease occurred in 42 patients (9.4%) at a rate of 89.4 per 1000 ECMO days and presented as acute cholecystitis in 33 patients (7.4%, 78.2/1000 ECMO days), acute cholangitis in 7 patients (1.6%, 159.1/1000 ECMO days), and biliary pancreatitis in 5 patients (1.1%, 151.5/1000 ECMO days). Of those, three patients showed both acute cholecystitis and cholangitis, and Mirizzi syndrome was not observed in any of the patients. Among the 33 patients with acute cholecystitis, four had acalculous cholecystitis due to GB hematoma (0.9%, 57.1/1000 ECMO days). During the same study period, 11,501 patients were treated without ECMO application in our ICU, of which 188 (1.6%) were diagnosed with complicated GB disease.

The most common presentation was both elevated LFT levels and hyperbilirubinemia (52 patients, 83.9%), followed by septic shock (40 patients, 64.5%). Nausea and vomiting were observed in 23 patients (37.1%). Only 14 patients (22.6%) presented with right upper quadrant abdominal pain and 13 patients (21%) with fever. Treatment for GB disease was performed only in complicated GB disease and included the following: percutaneous transhepatic gallbladder drainage (PTGBD, *n* = 33, 53.2%), followed by percutaneous transhepatic biliary drainage (PTBD, *n* = 7, 11.3%). Among these, nine patients underwent laparoscopic cholecystectomy after PTGBD. All nine cases of cholecystectomy were performed laparoscopically after the patient was successfully weaned from ECMO and transferred to the general ward.

### 3.2. Patient Characteristics

There were significant statistical differences in the age and proportion of elderly patients (≥65 years) between the two groups (Table 1). The average age of patients in the G group was 62 (54.8–69.3) years, which was greater than that of patients in the N group (57 (47.3–66) years, *p* = 0.006), and there was a significantly higher proportion of elderly patients in the G group (43.5% vs. 27.1%, *p* = 0.008). With regard to risk factors for GB disease, the proportion and duration of total parenteral nutrition (TPN) were significantly different between the two groups. The proportion of TPN use in the G group was 64.5%, which was higher than that in the N group (47.4%), and the duration of TPN (median (IQR), 3 (0–9.3) vs. 0 (0–4) days, *p* = 0.001) was significantly longer in the G group than in the N group (Table 1). In addition, the duration of fasting tended to be longer in the G group than in the N group (median (IQR), 3.5 (2–6.3) vs. 3 (1–5), *p* = 0.055).

### 3.3. Risk Factors for GB Disease during ECMO Therapy

In univariate Cox proportional hazards regression analysis, the durations of nil per os (>7 days), elevated plasma Hb (>50 mg/dL), and longer ECMO support (>2 weeks) were significantly associated with GB disease during ECMO therapy. In multivariate analysis, only longer ECMO support (>2 weeks) (hazard ratio (HR), 2.95, 95% confidence interval (CI), 1.69–5.15, *p* < 0.001) and elevated plasma Hb (>50 mg/dL) (HR, 2.12, 95% CI, 1.18–3.78, *p* = 0.012) remained significant (Table 2). The occurrence of GB disease according to the ECMO period and plasma Hb level is also shown in Figure 2 and Figure 3. As the period of ECMO therapy extended to more than one week, the risk of developing GB disease also continued to increase. Compared with those with an ECMO period of less than 1 week, the risk of developing GB disease was 16.88 times higher in those who received ECMO therapy for longer than 3 weeks (*p* < 0.001, Figure 2). Compared with those with a normal peak plasma Hb level (reference range, ≤11 mg/dL), the risk of developing GB disease was 24 times higher in patients with plasma Hb > 50 mg/dL and <100 mg/dL (*p* = 0.002), and 55.42 times higher in patients with plasma Hb > 100 mg/dL (*p* < 0.001, Figure 3).

### 3.4. Impact of GB Disease during ECMO Therapy on Clinical Outcomes

To reduce bias from confounding variables, we conducted propensity score matching analyses (1:2) between the two groups, adjusting for age and severity, using measures such as the SOFA and APACHE II scores. The G group (*n* = 62) and N group (*n* = 124) were adequately balanced after 1:2 propensity matching (Table 3).

There was no significant difference in ECMO duration between the two groups (7 (4–16.3) vs. 8 (4–14), *p* = 0.835). The proportion of long-term ECMO support (≥2 weeks) was not significantly different between the two groups (30.6% vs. 25.8%, *p* = 0.486). The duration of mechanical ventilation was significantly longer in the G group than in the N group (29.5 (11.6–51.2) vs. 17.0 (7.2–35.8) days, *p* = 0.013). In addition, the ICU and hospital survival rates were significantly lower in the G group than in the N group (ICU survival rate, 64.5% vs. 84.7%, *p* = 0.002; hospital survival rate, 59.7% vs. 81.5%, *p* = 0.001) (Table 4).

The incidence of other ECMO complications is shown in Appendix A. The proportions of hyperbilirubinemia (62.9% vs. 43.5%, *p* = 0.013), severe hemolysis (plasma Hb > 100 mg/dL) (30.6% vs. 2.4%, *p* < 0.001), and retroperitoneal bleeding (4.8% vs. 0, *p* < 0.001) were significantly higher in the G group than in the N group.

## 4. Discussion

In this study, we investigated the prevalence, related factors, and clinical outcomes of GB disease developed during ECMO therapy. Symptomatic GB disease occurred in 13.9% of patients treated with ECMO (76.2/1000 ECMO days), and the incidence was higher than that in the general ICU patients (1%), as also reported in other studies [12,13]. Acute acalculous cholecystitis accounts for approximately 10% of all acute cholecystitis cases, and its incidence was also higher in patients receiving ECMO than that of general ICU patients (0.2% to 0.4%) [9,14,15,16]. High plasma Hb and a long duration of ECMO treatment were significantly associated with the development of GB disease during ECMO therapy (Figure 2 and Figure 3). GB disease during ECMO was significantly associated with poor clinical outcomes, such as longer ventilator support and increased mortality.

Despite the improvement in ECMO systems and management, several complications remain, which affect patients’ survival and quality of life. To our knowledge, there has been no comprehensive analysis of the incidence and risk factors for the development of GB disease in patients receiving ECMO therapy. In this study, GB disease was a common complication for patients receiving ECMO therapy and was associated with higher mortality. The administration of ECMO is accompanied by many factors that can contribute to the development of the GB disease, such as ischemia–reperfusion injury, pro-inflammatory response, oxidative tissue stress, bile stasis, opioid therapy, positive pressure ventilation, and parenteral nutrition [17,18,19]. Furthermore, there are additional risk factors, including technical-induced hemolysis by ECMO [20]. Previously, the pathogenesis of hemolysis-induced gallstones was studied in hemolytic anemia [21]. Hemolysis leads to increased bilirubin excretion and pigment gallstone formation, which can lead to the development of gallstones.

Hemolysis is commonly observed and is associated with a number of adverse outcomes in patients receiving ECMO [21]. In this study, severe hemolysis (plasma Hb > 100 mg/dL) was observed in 7% (*n* = 31) of patients, and the incidence was comparable to the value of 5–18% reported previously [22]. The risk of GB disease was significantly increased when the peak plasma Hb level was ≥50 mg/dL or more and was 55 times greater for those with a peak plasma Hb level of >100 mg/dL, compared with those with a normal plasma Hb level (Figure 3). Among patients with plasma Hb levels greater than >100 mg/dL, 61.3% showed GB disease, and 54.8% showed complicated GB disease during ECMO therapy. Hemolysis during ECMO treatment can result from several factors, including the ECMO instrument’s circuit components and pre-ECMO patient characteristics [23]. It is more commonly observed when ECMO is administered over a prolonged period or under a relatively lower or higher flow [21]. As part of this process, Hb is broken down, the heme component is degraded to bilirubin in the liver, and bile is supersaturated with calcium bilirubinate to form biliary sludge or a pigmented stone [24,25,26,27]. Therefore, in addition to ischemic biliary injury caused by ECMO application, hemolysis may contribute to hyperbilirubinemia, increased bilirubin excretion, and gallstone formation. Hence, it is necessary to pay attention to the development of hemolysis in ECMO patients, and in patients undergoing extended ECMO support, more attention should be paid to the development of GB disease.

The diagnosis of complicated GB disease in ECMO-treated patients is challenging. In this study, LFT elevation was the most common presentation, and abdominal pain or fever was uncommon. Since leukocytosis or jaundice are not specific findings in ECMO-treated patients, and the results of laboratory evaluations may not be a reliable method of diagnosis [28]. Therefore, if a patient being treated with ECMO shows signs of persistent hypotension and hypoperfusion, GB disease should be considered. Bedside ultrasound can assist in the diagnosis of complicated GB disease in patients on ECMO [13]. CT is as accurate for the diagnosis of acute acalculous cholecystitis as ultrasound; however, ultrasound is the preferred diagnostic method because it can be performed quickly and conveniently at the bedside.

Generally, GB disease can lead to poor clinical outcomes in critically ill patients [9,19,29,30,31]. In this study, the ICU survival rate and hospital survival rate were significantly lower in the G group compared with the N group after propensity matching. The duration of ventilator support was significantly longer in patients with GB disease than in patients without GB disease. Generally, early cholecystectomy is the preferred treatment for GB disease [32,33]. However, cholecystectomy is not viable for many patients receiving ECMO. Instead, PTBD or PTGBD can be considered as an alternative treatment option for these patients [34]. In this study, PTGBD was conducted in 53.2% of the patients (33 patients), and cholecystectomy was performed for 27.3% (nine patients). There were no complications or deaths associated with the treatment of GB disease.

This study has several limitations owing to its single-center design and retrospective nature. First, it may be difficult to apply these results to other institutions as the results may not be generalizable to patients of different races and ages [35]. Second, prospective ultrasound screening for GB disease could not be performed in the overall population. Therefore, the number of asymptomatic cases may have been underestimated, or cases of pre-existing GB disease may have been overestimated.

In conclusion, this is the first study to investigate the incidence and possible causes of GB disease in a large cohort of adult patients treated with ECMO. ECMO-induced hemolysis should be monitored by checking the level of daily plasma Hb during routine care and screening, as well as when hemolysis is first suspected. Clinicians should recognize possible circuit malfunctions or improper flow settings. Medical efforts should be made to wean the patient from ECMO as early as possible. We also suggest that clinicians should actively consider GB disease and perform ultrasonography for patients receiving ECMO, especially patients with extended ECMO support and elevated plasma Hb, or in the presence of hypotension, hypoperfusion, and multi-organ failure.

## Figures and Tables

**Figure 1 jcm-11-02199-f001:**
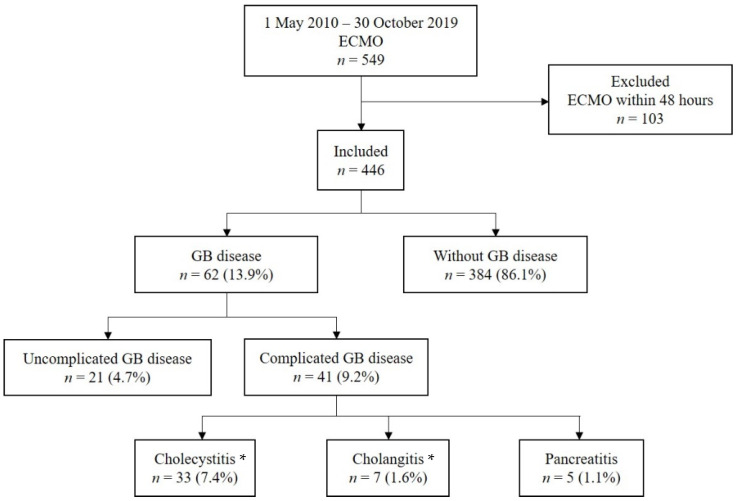
Consort flow diagram of the study enrolment. A total of 13.9% of patients were diagnosed with gallbladder (GB) disease. Among them, 9.4% developed complicated GB disease. * A total of 3 patients showed both acute cholecystitis and cholangitis.

**Figure 2 jcm-11-02199-f002:**
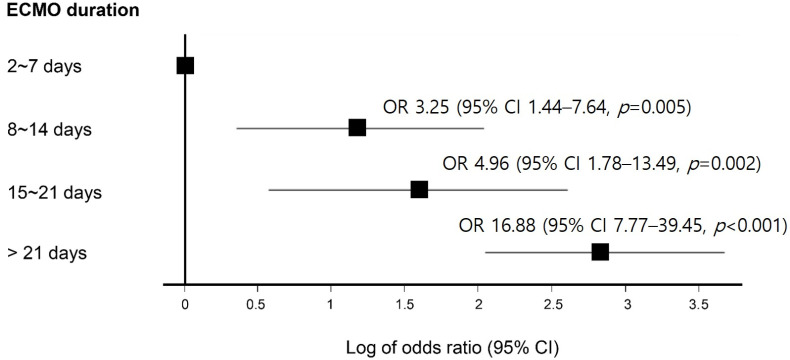
The effect of extracorporeal membrane oxygenation therapy duration on gallbladder disease. The risk of GB disease according to ECMO duration was expressed as a figure using a forest plot. To improve the readability of the figure, the odds ratio (OR) forest plot was plotted as log OR. As the period of ECMO therapy extended to more than one week, the risk of developing GB disease also continued to increase.

**Figure 3 jcm-11-02199-f003:**
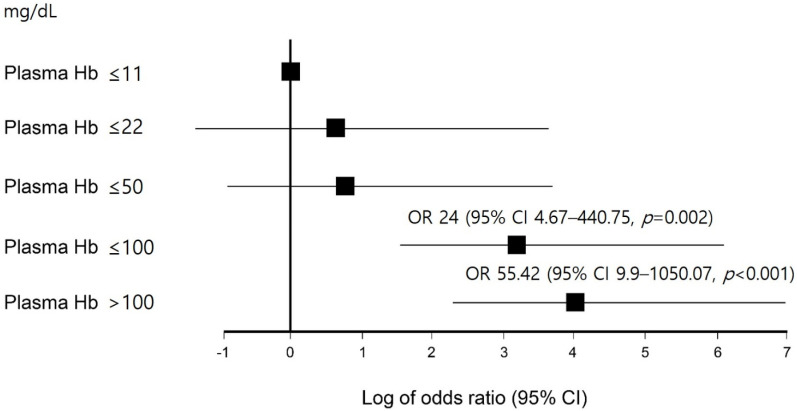
The effect of plasma hemoglobin on gallbladder disease. To improve the readability of the figure, the odds ratio (OR) forest plot was plotted as log OR. The reference ranges for plasma hemoglobin concentrations are lower than 11 mg/dL.

**Table 1 jcm-11-02199-t001:** Patients’ characteristics.

Variables	G Group (*n* = 62)	N Group (*n* = 384)	*p*
Age (years)	62 (54.8–69.3)	57 (47.3–66)	0.006
Elderly	27 (43.5)	104 (27.1)	0.008
Male	39 (62.9)	228 (59.4)	0.599
BMI (kg/m^2^)	22.5 (19.9–25.1)	23.0 (20.8–25.6)	0.300
APACHE II	11.5 (9–16.3)	13 (9–17)	0.601
SOFA score	12 (9–13)	11 (8–13)	0.256
Vasopressor	43 (69.4)	283 (73.7)	0.474
CRRT	36 (58.1)	184 (47.9)	0.138
Indication of ECMO			0.965
Respiratory	34 (54.8)	209 (54.4)	
Cardiac	19 (30.6)	123 (32.0)	
E-CPR	9(14.5)	52 (13.5)	
ECMO mode			0.502
VV	34 (54.8)	180 (46.9)	
VA	25 (40.3)	180 (46.9)	
VVA	0	7(1.8)	
other	3(4.8)	17 (4.4)	
Mechanical ventilator	62 (100)	374 (97.4)	0.199
Risk factors of GB disease			
Diuretics	54 (87.1)	304 (79.2)	0.145
LC	7 (11.3)	26 (6.8)	0.207
TPN	40 (64.5)	182 (47.4)	0.012
Duration of TPN (days)	3 (0–9.3)	0 (0–4)	0.001
Fasting	56 (90.3)	350 (91.1)	0.833
Duration of fasting (days)	3.5 (2–6.3)	3 (1–5)	0.055
Total bilirubin before ECMO	0.6 (0.4–1.3)	0.6 (0.3–1.1)	0.476

APACHE, acute physiology and chronic health evaluation; BMI, body mass index; CRP, C-reactive protein; CRRT, continuous renal replacement therapy; ECMO, extracorporeal membrane oxygenation; GB, gallbladder; G group, a group with GB disease; LC, liver cirrhosis; N group, a group without GB disease; SOFA, sequential organ failure assessment score; TPN, total parenteral nutrition; VA, venoarterial; VV, venovenous; VVA, venovenous arterial. Data are presented as median (interquartile range, IQR), or *n* (%).

**Table 2 jcm-11-02199-t002:** Cox regression analysis for gallbladder disease during extracorporeal membrane oxygenation treatment.

	Univariate		Multivariate	
Variables	HR (95% CI)	*p*	HR (95% CI)	*p*
Long-term ECMO	3.34 (1.91–5.82)	<0.001	2.95 (1.69–5.15)	<0.001
Plasma Hb > 50 mg/dL	2.43 (1.39–4.25)	0.002	2.12 (1.18–3.78)	0.012
NPO > 7 days	2.75 (1.53–4.96)	0.001		

ECMO, extracorporeal membrane oxygenation; Hb, hemoglobin; HR, hazard ratio; CI, confidence interval; NPO, nil per os.

**Table 3 jcm-11-02199-t003:** Patient characteristics in the propensity-matched cohort.

Variables	G Group (*n* = 62)	N Group (*n* = 124)	*p*
Age (years)	62 (54.8–69.3)	59 (50–67)	0.120
Elderly	27 (43.5)	41 (33.1)	0.162
Male	39 (62.9)	75 (60.5)	0.749
BMI (kg/m^2^)	22.5 (19.9–25.1)	22.2 (19.8–25.8)	1.000
APACHE II	11.5 (9–16.3)	12 (9–17)	0.756
SOFA score	12 (9–13)	12 (9–15.8)	0.832
Vasopressor	43 (69.4)	94 (75.8)	0.346
CRRT	36 (58.1)	68 (54.8)	0.676
Indication of ECMO			0.391
Respiratory	34 (54.8)	73 (58.9)	
Cardiac	19 (30.6)	41 (33.1)	
E-CPR	9 (14.5)	10 (8.1)	
ECMO mode			0.460
VV	34 (54.8)	59 (47.6)	
VA	25 (40.3)	54 (43.5)	
VVA	0	4 (3.2)	
other	3 (4.8)	7 (5.6)	
Mechanical ventilator	62 (100)	124 (100)	
Risk factors of GB disease			
Diuretics	54 (87.1)	102 (82.3)	0.398
LC	7 (11.3)	10 (8.1)	0.472
TPN	40 (64.5)	68 (54.8)	0.207
Duration of TPN (days)	3 (0–9.3)	2 (0–6.8)	0.464
Fasting	56 (90.3)	116 (93.5)	0.432
Duration of fasting (days)	3.5 (2–6.3)	3.5 (1–7)	1.000
Total bilirubin before ECMO	0.6 (0.4–1.3)	0.7 (0.3–1.6)	0.836

APACHE II, acute physiology and chronic health evaluation II; BMI, body mass index; CRP, C-reactive protein; CRRT, continuous renal replacement therapy; ECMO, extracorporeal membrane oxygenation; GB, gallbladder; G group, a group with GB disease; LC, liver cirrhosis; N group, a group without GB disease; SOFA, sequential organ failure assessment score; TPN, total parenteral nutrition; VA, venoarterial; VV, venovenous; VVA, venovenous arterial. Data are presented as median (interquartile range, IQR), or *n* (%).

**Table 4 jcm-11-02199-t004:** Clinical outcomes in the propensity-matched cohort.

Variables	G Group (*n* = 62)	N Group (*n* = 124)	*p*
Duration of ECMO, days	7 (4–16.3)	8 (4–14)	0.835
Long-term ECMO (≥2 weeks)	19 (30.6)	32 (25.8)	0.486
Duration of mechanical ventilation, days	29.5 (11.6–51.2)	17.0 (7.2–35.8)	0.013
Length of ICU stay, days	32.1 (17–52)	23.5 (11–43.7)	0.120
ICU survival	40 (64.5)	105 (84.7)	0.002
Survival discharge	37 (59.7)	101 (81.5)	0.001

ECMO, extracorporeal membrane oxygenation; G group, a group with GB disease; N group, a group without GB disease; ICU, intensive care unit. Data are presented as median (interquartile range, IQR), or *n* (%).

## Data Availability

The data that support the findings of this study are available from the corresponding author upon reasonable request.

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
