# Peer review of "Overlooked but Serious Gallbladder Disease during Extracorporeal Membrane Oxygenation: A Retrospective Analysis"

_jcm, 2022, doi:10.3390/jcm11082199_

Round 1
Reviewer 1 Report
The authors conducted a retrospective study to investigate the prevalence, related factors, and clinical outcomes of gallbladder disease during extracorporeal membrane oxygenation. This is an interesting and well-written study that discusses an important subject. The results of this study would certainly be of interest in clinical practice. I just have a few comments.
- Elevated plasma Hb and prolonged extracorporeal membrane oxygenation therapy were significant factors in the development of gallbladder disease. How to manage and prevent gallbladder disease in such patients? Furthermore, from the results of this study, do you have any advice on the medical care of patients during extracorporeal membrane oxygenation? Please answer this in the ‘Discussion’ section.
Author Response
- Elevated plasma Hb and prolonged extracorporeal membrane oxygenation therapy were significant factors in the development of gallbladder disease. How to manage and prevent gallbladder disease in such patients? Furthermore, from the results of this study, do you have any advice on the medical care of patients during extracorporeal membrane oxygenation? Please answer this in the ‘Discussion’ section.
Thank you for your comment. ECMO-induced hemolysis should be monitored by checking the level of daily plasma Hb during routine care and screening, as well as when hemolysis is first suspected. Clinicians should recognize possible circuit malfunctions or improper flow settings. Medical efforts should be made to wean the patient from ECMO as early as possible. I added this contents in the Discussion (Page 9, Line 286-90).
We suggest that clinicians should actively consider GB disease and perform ultrasonography for patients receiving ECMO, especially patients with extended ECMO support and, elevated plasma Hb, or in the presence of hypotension, hypoperfusion, and multi-organ failure. I already describe these in the end of the Discussion (Page 9, Line 290-3).
Our manuscript was edited by https://www.mdpi.com/authors/english.

Reviewer 2 Report
Comments to the Authors
The aim of the study was to investigate the prevalence, related factors and clinical outcomes of gallbladder disease in patients treated with ECMO. The manuscript is well written and results are clearly reported, depicting the clinical relevance of the paper. Only minor reviews are suggested.
Minor comments
- When referring to the 11,501 patients treated in your ICU, are the ECMO patients part of those? Please clarify.
- Nine patients underwent cholecystectomy after percutaneous transhepatic biliary drainage (PTBD). What was the timing of cholecystectomy after PTBD? How many procedures were open vs. laparoscpic.
- What is the definition of “elderly patients”?
Author Response
1. When referring to the 11,501 patients treated in your ICU, are the ECMO patients part of those? Please clarify.
Thank you for your comment. Eleven thousands, and 501 patients were treated without ECMO application in our ICU. I added this contents in the Result (Page 3, Line 123-4).
2. Nine patients underwent cholecystectomy after percutaneous transhepatic biliary drainage (PTBD). What was the timing of cholecystectomy after PTBD? How many procedures were open vs. laparoscpic.
Thank you for your comment. All nine cases of cholecystectomy were performed laparoscopically after the patient was successfully weaned from ECMO, and transferred to the general ward. I described this contents in the Result (Page 4, Line 137-9).
3. What is the definition of “elderly patients”?
Thank you for your comment. Elderly patients is defined as over 65 years of age. I already described this in the Result (Page 4, Line 140).
Our manuscript was edited by https://www.mdpi.com/authors/english.
